# Adenoid Cystic Carcinoma of the Breast May Be Exempt from Adjuvant Chemotherapy

**DOI:** 10.3390/jcm11154477

**Published:** 2022-07-31

**Authors:** Lixi Li, Di Zhang, Fei Ma

**Affiliations:** Department of Medical Oncology, National Cancer Center/National Clinical Research Center for Cancer/Cancer Hospital, Chinese Academy of Medical Sciences and Peking Union Medical College, Chaoyang District, Panjiayuan Nan Road 17, Beijing 100021, China; drlilixi@student.pumc.edu.cn (L.L.); Iris0208@student.pumc.edu.cn (D.Z.)

**Keywords:** adenoid cystic carcinoma of the breast, adjuvant chemotherapy, survival, prognosis, surveillance, epidemiology, and end results database

## Abstract

Consistent standards regarding whether postoperative adjuvant chemotherapy is required in the treatment of adenoid cystic carcinoma of the breast (ACCB) are currently lacking. Using clinical data from the Surveillance, Epidemiology, and End Results (SEER) database (1988–2015), and the National Cancer Center of China (2004–2020), we retrospectively analyzed patients with ACCB who received radical treatment. A total of 661 patients were eligible. The median age at diagnosis was 61 years; 99.5% of patients were initially diagnosed with stage I and II breast cancer, and 76.7% had triple-negative breast cancer. Only 12.4% of patients received adjuvant chemotherapy. Multivariate analysis showed that patients with lymph node metastasis and non-radiotherapy had worse overall survival (OS) (*p* < 0.05). Patients with lymph node metastasis, stage IIB and III, histological grade ≥ 2, and non-radiotherapy had worse breast cancer-specific survival (BCSS) (*p* < 0.05). Adjuvant chemotherapy did not improve the OS or BCSS. Patients treated with adjuvant chemotherapy also had no better survival outcomes after propensity score matching. External data verification confirmed that chemotherapy did not improve disease-free survival or OS. Adjuvant chemotherapy cannot improve the clinical outcomes of ACCB, even in subgroups with a high risk of recurrence and metastasis.

## 1. Introduction

Adenoid cystic carcinoma (ACC) usually occurs in the salivary glands but is relatively rare in the breast, prostate, esophagus, trachea, and other sites [1,2,3,4]. Adenoid cystic carcinoma of the breast (ACCB) accounts for approximately 0.06% to 0.1% of all breast cancers [2,5]. The molecular subtype of ACCB is mainly triple-negative breast cancer (TNBC), but its clinical features and prognosis are quite different from those of other pathological types of TNBC [6]. It is a slow-growing tumor characterized by rare lymph node involvement and favorable prognosis, with a 5-year overall survival (OS) rate of 98–100% [2,7,8]. To date, there is no standard treatment for ACCB. Surgery is the primary treatment strategy followed by adjuvant radiotherapy [9]. However, consistent standards regarding whether adjuvant chemotherapy is required are lacking.

Therefore, we conducted a population-based study to explore the prognostic significance of adjuvant chemotherapy using the Surveillance, Epidemiology, and End Results (SEER) program database. We simultaneously used data from our center for external verification.

## 2. Materials and Methods

### 2.1. Data Collection

Patients who were initially diagnosed with ACCB between 1 January 1988 and 31 December 2015 were retrieved from the SEER database. The inclusion criteria were as follows: (1) cases of ACCB with histopathological confirmation; (2) patients receiving radical surgery; (3) adjusted American Joint Committee on Cancer (AJCC) 6th ed. stage I–III. We excluded cases of unperformed or unknown surgery, patients with duplicate patient ID (inconsistent information), patients with unknown TNM staging or stage IV, and those with unknown estrogen receptor (ER) or progesterone receptor (PR) status. The patient screening process is illustrated in Figure 1. The clinicopathological characteristics analyzed were age at diagnosis, sex, TNM staging, tumor size, lymph node, ER and PR status, histological grades, and treatment modalities (including radiotherapy/chemotherapy).

In addition, we collected clinical data of patients with ACCB at the National Cancer Center of China from January 2004 to December 2020 and used them for external verification. The inclusion criteria were the same as before, and we excluded patients with incomplete clinicopathological and treatment information. This study was approved by the ethics committee of the Chinese Academy of Medical Sciences and Peking Union Medical College.

### 2.2. Statistical Analysis

OS was the interval from the diagnosis of ACCB to death from any cause or the last follow-up. Breast cancer-specific survival (BCSS) was calculated as the time from the initial diagnosis to the date of breast cancer-specific death. Disease-free survival (DFS) was calculated from the beginning of radical surgery to the recurrence of the disease or death from any cause. The classification data are expressed as frequency and percentage, and survival data are presented as median and survival rates. We compared variables between groups using Pearson’s chi-square and Fisher’s exact probability tests (two-sided). The Kaplan–Meier method was used to describe the survival curve, and the survival analysis and OS and BCSS rates were calculated. We used the log-rank test to compare the differences in survival curves between the two groups. Multivariate Cox analysis was performed using stepwise forward regression analysis to evaluate independent prognostic factors. Statistical analyses were performed using the statistical package for the social sciences (SPSS) software (version 24.0, IBM, Armonk, NY, USA), and figures were produced using GraphPad Prism (version 9.0, GraphPad Software Inc., San Diego, CA, USA). To eliminate the obvious differences in baseline covariates and inherent selection bias, we conducted propensity score matching (PSM) analysis between the patients in chemotherapy and non-chemotherapy groups and performed forest plots using the R software (version 3.6.4, R Foundation, Vienna, Austria). *p* < 0.05 was considered statistically significant for all analyses.

## 3. Results

### 3.1. Clinicopathological Characteristics

We enrolled 661 patients with ACCB who underwent radical surgery (Figure 1), the baseline clinical characteristics of which are shown in Table 1. The median age at diagnosis was 61 years. Of the patients enrolled, 99.5% (658) were diagnosed with stage I and stage II breast cancer, and only 3 cases were stage III. The positive ratio of ER and PR status accounted for 20.7% and 12.9%, respectively. Among the 249 patients with available human epidermal growth factor 2 receptor (HER2) data, only 3 (1.2%) were positive, and 76.7% (191/249) of the patients with a clear molecular subtype were TNBC. Only 82 (12.4%) of the 661 patients received postoperative adjuvant chemotherapy. The results showed that patients younger than 60 years at diagnosis, with advanced TNM stage (IIB and III), tumor size > 2 cm, lymph node metastasis, and high histological grade (≥grade 2), and who were PR-positive were more likely to receive chemotherapy.

### 3.2. Prognostic Factors

The 5- and 10-year OS rates for patients with ACCB were 88.4% and 75.0%, respectively. In addition, the 5- and 10-year BCSS rates were 95.8% and 92.9%, respectively (Figure 2).

Univariate analysis showed that patients aged ≥60 years at diagnosis, with lymph node metastasis, stage IIB and III, and histological grade ≥2, and patients not receiving postoperative radiotherapy had worse OS (*p* < 0.05) (Table 2). Patients with tumors >2 cm, lymph node metastasis, advanced stage, high histological grade, and patients not receiving radiotherapy had worse BCSS (*p* < 0.05) (Figure 2).

The prognostic factors were further explored using multivariate Cox regression analysis, which suggested that patients with lymph node metastasis and those who did not receive postoperative radiotherapy had poorer OS (*p* < 0.05). Patients with large tumor size, lymph node metastasis, advanced stage, high histological grade, and patients not receiving radiotherapy had worse BCSS (*p* < 0.05). Moreover, postoperative adjuvant chemotherapy did not significantly improve OS or BCSS (*p* > 0.05) (Table 2).

### 3.3. Propensity Matching Score and Subgroup Analysis

To explore the subgroups of patients who benefited from chemotherapy, we performed PSM for patients who received and did not receive chemotherapy. Through neighbor matching, no statistically significant difference was found between the chemotherapy and non-chemotherapy groups in terms of age at diagnosis, TNM stage, tumor size, lymph node metastasis status, ER status, PR status, and histological grade (Table 3). There was no statistically significant difference in OS and BCSS between these groups (OS: *p* = 0.455, BCSS: *p* = 0.148). The distribution graphs and histograms of propensity scores before and after matching are listed in Appendix A. The results are shown as forest plots of hazard ratios (HRs) and 95% confidence intervals (CIs) for OS and BCSS (Figure 3 and Figure 4). In the subgroup aged ≥60 years, chemotherapy prolonged OS (*p* = 0.038, HR: 0.454, 95% CI: 0.216–0.956). In the subgroup aged <60 years, the BCSS of the chemotherapy group was worse (*p* = 0.041, HR: 4.956, 95% CI: 1.07–22.96).

### 3.4. Clinical Practice in China

To better understand the status of medical treatment and prognosis for patients with ACCB between Chinese and international clinical practice, we summarized the clinical characteristics and treatment modalities of patients with ACCB in the National Cancer Center in China. We included 26 patients with ACCB who underwent surgery, and of these, 7 patients received postoperative adjuvant chemotherapy. The postoperative chemotherapy regimens included epirubicin combined with cyclophosphamide, paclitaxel combined with cyclophosphamide, docetaxel combined with cyclophosphamide, and cyclophosphamide combined with mycin and 5-fluorouracil. There was no statistically significant difference between the chemotherapy and non-chemotherapy groups in terms of baseline clinical characteristics (Appendix A). Survival analysis showed that postoperative adjuvant chemotherapy did not improve the OS in patients with ACCB (χ^2^ = 0.476, *p* = 0.490). Similarly, postoperative adjuvant chemotherapy did not improve DFS (χ^2^ = 0.966, *p* = 0.326) (Appendix A).

## 4. Discussion

To our knowledge, this is the first study that systematically discusses the clinical features and prognostic differences between ACCB patients receiving chemotherapy and those not.

ACCB is a rare subtype of TNBC with a relatively indolent disease course. It is diagnosed at an earlier stage and has less regional lymph node metastasis. In our study, approximately 80% of patients had no lymph node metastasis, and >50% of the patients had a tumor size ≤2 cm, which was consistent with the results reported in previous studies [10]. In addition, most patients in our analysis were mainly stage I–II and had a well-differentiated grade. The ACCB conferred an excellent prognosis, with 5-year BCSS and OS rates of 95.8% and 88.4%, respectively. Similar to other pathological types of breast cancer, patients with ACCB with tumor size >2 cm, lymph node metastasis, and histological grade ≥2 had a poor prognosis. Our results were consistent with those of some studies [9,11], which showed that postoperative adjuvant radiotherapy improves BCSS and OS in patients with ACCB.

TNBC often has highly aggressive clinical features and is insensitive to endocrine and targeted therapies due to the loss of hormone receptors and HER2 expression. Surgery combined with adjuvant chemotherapy has been recommended as the preferred treatment for most patients with TNBC, and recent clinical trial results have confirmed that maintaining capecitabine and olaparib after surgery can prolong DFS. Most cases of ACCB are of the TNBC subtype; however, ACCB demonstrated favorable characteristics, including older age at diagnosis, good differentiation, lymph node negativity, and small tumor size [10,12]. To date, the primary treatment for ACCB is surgery, and postoperative follow-up is necessary because few patients are still prone to local recurrence and distant metastasis after surgery [13,14]. There are cases of lung, liver, and bone metastases in ACCB, which can adversely affect long-term survival [15]. In addition, ACCB can be transformed into high-level TNBC through clonal selection and gene modification [16]; therefore, optimizing the management of ACCB should be further explored. Currently, the role of adjuvant chemotherapy is still unclear, and its benefit to the survival of patients with ACCB has not been well evaluated. Generally, our study presented the characteristics and survival outcomes of ACCB and comprehensively compared the differences in the features and prognosis between adjuvant chemotherapy and non-chemotherapy.

Few studies have evaluated the value of adjuvant chemotherapy in ACCB, and the guidelines for its management are not well established. Data from the National Cancer Center in the United States indicated that only 12.9% of patients with ACCB received adjuvant chemotherapy ^6^, and our results were close to these findings at 12.4% (7/26). Strikingly, we collected a larger sample size from the SEER database and the National Cancer Center in China, providing a more complete and extensive analysis for ACCB and yielding more robust and reliable information. The results of the multivariate analysis further suggested that adjuvant chemotherapy was not helpful in the clinical outcome of ACCB. The BCSS and OS of patients who received postoperative adjuvant chemotherapy were shorter than those of patients who did not receive adjuvant chemotherapy. After PSM, subgroup analysis showed that chemotherapy did not improve OS and BCSS regardless of tumor size, lymph node metastasis status, AJCC TNM stage, ER, and PR status. Among patients aged <60 years, receiving chemotherapy had a poor BCSS. In patients aged ≥60 years, the results showed that chemotherapy prolonged OS, but BCSS did not improve. This may be related to the better performance status and fewer underlying diseases in patients who received chemotherapy. It should be emphasized that the results of the subgroup analysis of BCSS showed that in certain subgroups, especially stage I and grade I subgroups, the HR value and 95% CI were too large due to too few end-point events. This also suggests that early and well-differentiated ACCBs are less aggressive, have a better prognosis, and are rarely associated with BCSS. Therefore, for patients with ACCB, including triple-negative ACCB, surgery and postoperative radiotherapy are the preferred treatment methods, and there is currently insufficient evidence-based medical evidence to recommend adjuvant chemotherapy for these patients. It should be emphasized that in some subgroup analyses, such as the stage IIb/III subgroup (only 28 patients), the statistical power is still lacking due to the small number of samples. However, we first proposed the clinical impact of adjuvant chemotherapy in ACCB, providing additional evidence for the management of this rare malignancy. Furthermore, a larger sample size of data is needed for validation in the future.

Adenoid cystic carcinoma predominantly occurs in the salivary glands, accounting for approximately 10–15% of salivary gland malignancies [17]. However, it is a very rare subtype in other malignancies such as prostate cancer and breast cancer [1,10].Patients with head and neck ACC are insensitive to chemotherapy and targeted therapy, and neither postoperative adjuvant chemotherapy nor palliative chemotherapy for advanced patients can improve patient prognosis [18,19]. After proper local treatment, chemotherapy only has a palliative effect on a small number of recurrent or metastatic ACC patients with clinical symptoms or disease progression [20,21]. Chemotherapy (whether single-drug or combination chemotherapy) is ineffective against most salivary gland ACCs [22]. However, there are some studies on ACCB chemotherapy. The results of a network meta-analysis showed that chemotherapy could not improve the OS and BCSS of HR+/LN- ACCB [23]. Using PSM, we further analyzed whether chemotherapy was beneficial in different subgroups and found that even among ACCB patients with axillary lymph node metastasis, tumor size >2 cm, histological grade >2, and HR negativity, chemotherapy did not improve OS and BCSS. Similar results found in the analysis of external data from the National Cancer Center of China indicated that chemotherapy could not improve DFS and OS. This means that despite more than 10% of patients receiving it, chemotherapy is not beneficial for ACCB patients. The above results imply that adjuvant chemotherapy indications recommended by the existing guidelines for the diagnosis and treatment of breast cancer are not suitable for ACCB, and the treatment strategy of ACCB cannot completely refer to TNBC. The favorable natural course of this particular histological type should be considered when deciding whether to administer adjuvant chemotherapy, and further verification of the specific beneficiaries and chemotherapy regimen of ACCB patients receiving adjuvant chemotherapy needs to be performed through prospective clinical trials.

Our study had several limitations. First, due to the retrospective nature of the analysis, selection bias existed even when PSM was used. In addition, specific chemotherapy regimens, which may have influenced the OS and BCSS, were not available from SEER. However, according to data from the National Cancer Center of China, patients with ACCB received chemotherapy regimens based on anthracyclines or taxanes. Future studies are needed to validate our results and further explore the chemotherapeutic sensitivity of ACCB to optimize the treatment of this rare disease. Furthermore, some specific and detailed data were not available in the SEER database, such as the site and dose of radiotherapy, and HER2 status (known only in 36% of patients), which makes it impossible for us to further analyze the relationship between these variables and ACCB.

## 5. Conclusions

In summary, tumor size, lymph node metastasis, AJCC stage, histological grade, and radiotherapy are important factors for the prognosis of patients with ACCB. The clinicopathological characteristics and treatment options of ACCB are different from those of invasive ductal carcinoma with TNBC subtype, and adjuvant chemotherapy cannot improve the survival of patients with ACCB. Even in subgroups with high-risk factors for recurrence and metastasis, such as axillary lymph node metastasis, tumor size >2 cm, and high histological grade, chemotherapy cannot improve OS and BCSS. Triple-negative breast cancer is a group of tumors with multiple pathological types and high heterogeneity, with the pathological types having different clinical features and prognoses. Molecular classification has important guiding value for the clinical diagnosis and treatment of breast cancer, but different pathological types of TNBC have different clinical diagnosis and treatment strategies. Comprehensive consideration and analysis should be carried out when formulating treatment strategies, and multidisciplinary consultations should be conducted when necessary.

## Figures and Tables

**Figure 1 jcm-11-04477-f001:**
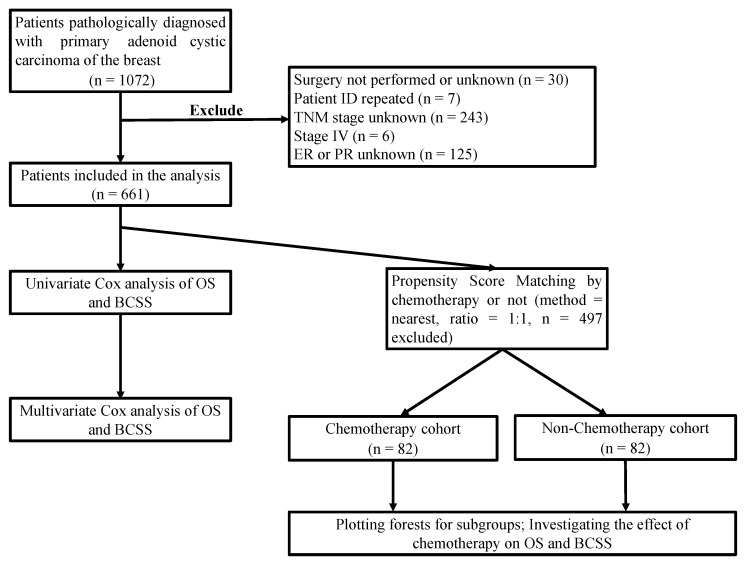
Recruitment of participants.

**Figure 2 jcm-11-04477-f002:**
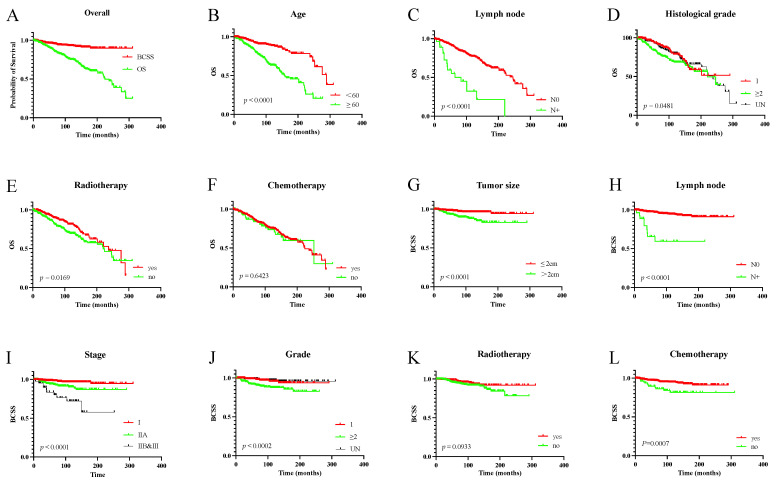
The OS and BCSS of overall population and different groups. (**A**): the OS and BCSS of the overall population; (**B**–**F**): the OS of different subgroups of age (**B**), lymph node (**C**), histological grade (**D**), radiotherapy (**E**), chemotherapy (**F**); (**G**–**L**): the BCSS of different subgroups of tumor size (**G**), lymph node (**H**), stage (**I**), grade (**J**), radiotherapy (**K**), chemotherapy (**L**).

**Figure 3 jcm-11-04477-f003:**
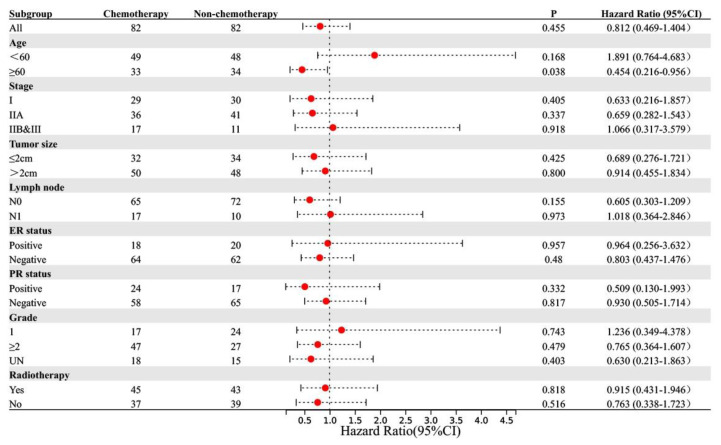
Subgroup analysis of OS in chemotherapy and non-chemotherapy groups.

**Figure 4 jcm-11-04477-f004:**
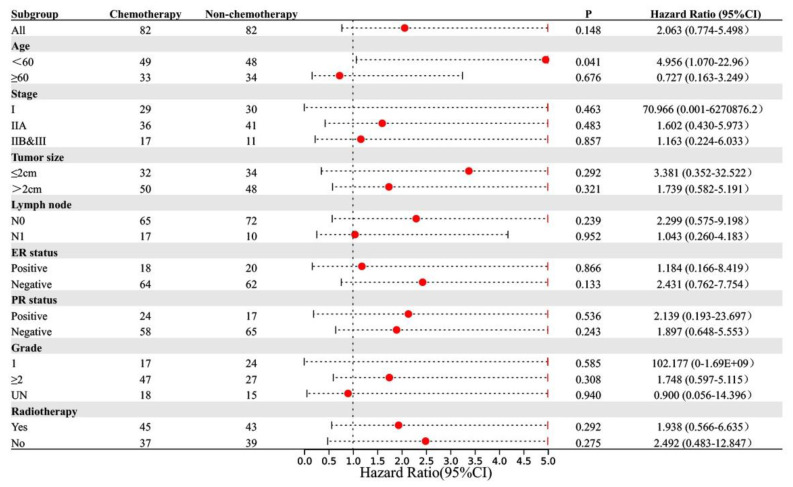
Subgroup analysis of BCSS in chemotherapy and non-chemotherapy groups.

**Table 1 jcm-11-04477-t001:** Clinical characteristics of patients with ACCB in chemotherapy group and non-chemotherapy group in SEER.

Characteristics	Total (n = 661)	Chemotherapy		χ^2^	*p*
		Yes (n = 82)	No (n = 579)		
Age				6.824	0.009
<60	306 (21.2)	49 (30.5)	257 (19.9)		
≥60	355 (78.8)	33 (69.5)	322 (80.1)		
Sex				-	1
Female	655 (99.1)	82 (100)	573 (99)		
Male	6 (0.9)	0 (0)	6 (1)		
Stage				40.453	<0.001
I	383 (57.9)	29 (35.4)	354 (61.1)		
IIA	236 (35.7)	36 (43.9)	200 (34.5)		
IIB and III	42 (5.9)	17 (17.1)	25 (4.3)		
Tumor size				15.444	<0.001
≤2 cm	390 (59.0)	32 (39)	358 (61.8)		
>2 cm	271 (41.0)	50 (61)	221 (38.2)		
Lymph node				-	<0.001
N0	634 (95.9)	65 (79.3)	569 (98.3)		
N1	27 (4.1)	17 (20.7)	10 (1.7)		
ER status				0.086	0.77
Positive	137 (20.7)	18 (22)	119 (20.6)		
Negative	524 (79.3)	64 (78)	460 (79.4)		
PR status				5.177	0.023
Positive	85 (12.9)	17 (20.7)	68 (11.7)		
Negative	576 (87.1)	65 (79.3)	511 (88.3)		
Grade				12.285	0.002
1	213 (32.2)	17 (20.7)	196 (33.9)		
≥2	263 (28)	47 (32.9)	216 (27.3)		
Unknown	185 (28)	18 (22)	167 (28.8)		
Radiotherapy				2.041	0.153
Yes	314 (47.5)	45 (54.9)	269 (46.5)		
No	347 (52.5)	37 (45.1)	310 (53.5)		

**Table 2 jcm-11-04477-t002:** Univariate and multivariate analyses of factors influencing OS and BCSS in ACCB.

Variable	OS				BCSS			
	Univariate		Multivariate		Univariate		Multivariate	
	HR (95% CI)	*p*	HR (95% CI)	*p*	HR (95% CI)	*p*	HR (95% CI)	*p*
Age								
<60 (ref)								
≥60	3.418 (2.354–4.961)	<0.001			1.08 (0.571–2.042)	0.814		
Stage								
I (ref)								
IIA	1.196 (0.849–1.686)	0.306			3.334 (1.538–7.226)	0.002		
IIB and III	2.071 (1.212–3.538)	0.008			10.24 (4.253–24.659)	<0.001		
Tumor size								
≤2cm (ref)								
>2 cm	1.173 (0.851–1.618)	0.33			3.527 (1.779–6.993)	<0.001	2.678 (1.332–5.386)	0.006
Lymph node								
N0 (ref)								
N1	4.689 (2.818-7.802)	<0.001	0.207 (0.125–0.345)	<0.001	9.756 (4.589–20.743)	<0.001	6.889 (3.126–15.18)	<0.001
ER status								
Positive (ref)								
Negative	1.089 (0.89–1.331)	0.408			1.506 (0.897–2.529)	0.121		
PR status								
Positive (ref)								
Negative	0.978 (0.626–1.528)	0.923			1.365 (0.757–2.462)	0.301		
Grade								
1 (ref)								
≥2	1.614 (1.077–2.418)	0.02			3.563 (1.471–8.633)	0.005	3.081 (1.256–7.56)	0.014
Unknown	1.152 (0.741–1.791)	0.529			0.827 (0.252–2.711)	0.754	0.792 (0.241–2.603)	0.7
Radiotherapy								
Yes	0.666 (0.48–0.923)	0.015	0.649 (0.467-0.901)	0.010	0.568 (0.290–1.11)	0.098	0.452 (0.228–0.897)	0.023
No (ref)								
Chemotherapy								
Yes	1.105 (0.703–1.736)	0.665			3.079 (1.553–6.105)	0.001		
No (ref)								

**Table 3 jcm-11-04477-t003:** Clinical characteristics of patients with ACCB in chemotherapy group and non-chemotherapy group after PSM in a 1:1 ratio.

Characteristics	Chemotherapy		χ^2^	*p*
	Yes (n = 82) (%)	No (n = 82) (%)		
Age			0.025	0.874
<60	49 (59.8)	48 (58.5)		
≥60	33 (40.2)	34 (41.5)		
Stage			1.627	0.443
I	29 (35.4)	30 (36.6)		
IIA	36 (43.9)	41 (50)		
IIB and III	17 (20.7)	11 (13.4)		
Tumor size			0.101	0.75
≤2 cm	32 (39)	34 (41.5)		
>2 cm	50 (61)	48 (58.5)		
Lymph node			2.172	0.14
N0	65 (79.3)	72 (87.8)		
N1	17 (20.7)	10 (12.2)		
ER status			0.137	0.711
Positive	18 (22)	20 (24.4)		
Negative	64 (78)	62 (75.6)		
PR status			1.593	0.207
Positive	24 (29.3)	17 (20.7)		
Negative	58 (70.7)	65 (79.3)		
Grade			0.922	0.631
1	17 (20.7)	24 (29.3)		
≥2	47 (57.3)	27 (32.9)		
Unknown	18 (22)	15 (18.3)		
Radiotherapy			0.098	0.754
Yes	45 (54.9)	43 (52.4)		
No	37 (45.1)	39 (47.6)		

## Data Availability

The data are available from the corresponding author on reasonable request.

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
