# Peer review of "Adenoid Cystic Carcinoma of the Breast May Be Exempt from Adjuvant Chemotherapy"

_jcm, 2022, doi:10.3390/jcm11154477_

Round 1

Reviewer 1 Report

The authors analyzed the impact of chemotherapy in ACCB patients using the SEER database.

It was really very interesting and enjoyable to read this article. By pure chance, I recently received a patient suffering from this rare tumor and this article allowed me to clarify some points.

It is an article that deserves to read in consideration of the scarce literature data, albeit with important methodological limitations.

Therefore I congratulate the authors and suggest some revisions:

Introduction and discussion:

ACC is a rare subtype that can occur in salivar gland, but not only. For exemple we can found it in prostate (Curr. Oncol. 2022, 29, 1866–1876. https://doi.org/10.3390/curroncol29030152).  Its should be mentionated.

The authors repeat several time that RT influences OS. It would be necessary, if possible, to describe the doses of RT and the techniques used.

Discussion: line 17

Authors says That the preferred  treatment is adjuvant chemotherapy after surgery. Why did you omit RT?

line 22: again, the author omitted RT. 

It should be expressed more clearly in the discussion (and in the abstract too) that it is not possible to derive data from chemotherap scheme and the percentage, as this is an important limitation of the study

Author Response

Reviewer Response

Comments and Suggestions for Authors

The authors analyzed the impact of chemotherapy in ACCB patients using the SEER database. It was really very interesting and enjoyable to read this article. By pure chance, I recently received a patient suffering from this rare tumor and this article allowed me to clarify some points. It is an article that deserves to read in consideration of the scarce literature data, albeit with important methodological limitations. Therefore, I congratulate the authors and suggest some revisions:

Response:

We thank for your positive evaluation of our work. You are a very experienced expert in this field, we deeply appreciate your valuable suggestions to improve the quality of our manuscript. Of course, we are also very eager to have the opportunity to get your further guidance on the new manuscript.

Introduction and discussion:

ACC is a rare subtype that can occur in salivar gland, but not only. For exemple we can found it in prostate (Curr. Oncol. 2022, 29, 1866–1876. https://doi.org/10.3390/curroncol29030152).  Its should be mentionated.

Response:

Thank you for raising this point, and we very much agree with you. In the revised manuscript, we have mentioned this point.

The authors repeat several time that RT influences OS. It would be necessary, if possible, to describe the doses of RT and the techniques used.

Response:

Thanks for your meaningful criticism. Regrettably, detailed information about radiotherapy (such as dose, radiation field, technique, etc.) is not available due to database opening restrictions, which was a pity. Because these limitations are inherent in secondary data analysis of this nature, we discuss this point adequately in the limitations section of the revised manuscript.

Discussion: line 17

Authors says That the preferred treatment is adjuvant chemotherapy after surgery. Why did you omit RT?

line 22: again, the author omitted RT.

Response:

Thanks for your meaningful criticism. Because our focus in this article is on adjuvant chemotherapy, there does exist a weakening of the description of radiotherapy. RT is also critical in the management for TNBC and this rare malignancy. Univariate and multivariate results showed that radiotherapy improved the prognosis of patients with ACCB, but we did not conduct further analysis of radiotherapy because the specific site and dose of radiotherapy were not available. We also discuss the limitations in the Discussion section of the revised manuscript.

It should be expressed more clearly in the discussion (and in the abstract too) that it is not possible to derive data from chemotherap scheme and the percentage, as this is an important limitation of the study

Response:

We deeply appreciate your valuable suggestions to improve our manuscript. As you suggested, in the revised manuscript, we have re-written some diction in the Discussion sections (and in the abstract too). Thanks again.

Reviewer 2 Report

In this study, the authors retrospectively analyzed patients with ACCB who received radical treatment. 12.4% of patients received adjuvant chemotherapy. Adjuvant chemotherapy did not improve the OS or BCSS. Patients treated with adjuvant chemotherapy also had no better survival outcomes after propensity score matching. External data verification confirmed that chemotherapy did not improve disease-free survival or OS. Based on the results, the authors concluded that adjuvant chemotherapy cannot improve the clinical outcomes of ACCB, even in subgroups with high risk of recurrence and metastasis.

Comment:

1.      Figure 2L, chemotherapy did show an improved survival (BCSS), but in the text, it stated that “postoperative adjuvant chemotherapy did not significantly improve OS or BCSS (Page 6). The authors need to justify this discrepancy.

2.      Table 1, the percentage for the subgrouping by tumor size (39% and 54.9%, 61.8% and 34.9%) don’t not add up to 100%. It needs to be fixed.

Author Response

Reviewer Response

English language and style are fine/minor spell check required 

Response:

Thanks for your meaningful criticism. We know that language polishing will make the paper easier to read and more fluent, and more in line with the writing habits of scientific English papers. Our paper has been polished by editage and we have provided Certificate of Editing to your journal.

Comments and Suggestions for Authors

Comment:

1.Figure 2L, chemotherapy did show an improved survival (BCSS), but in the text, it stated that “postoperative adjuvant chemotherapy did not significantly improve OS or BCSS (Page 6). The authors need to justify this discrepancy.

Response:

We deeply appreciate the reviewer’s suggestion. We agree with the comment and made some additions in revised manuscripts.

In our findings, univariate analysis showed that chemotherapy associated with BCSS (P < 0.05). However, in multivariate Cox regression analysis, adjusting for other variables, postoperative adjuvant chemotherapy did not significantly improve OS or BCSS (P > 0.05). Figure 2 shows the results of univariate analysis, while "postoperative adjuvant chemotherapy did not significantly improve OS or BCSS (Page 6)" is the result of multivariate analysis, corresponding to the multivariate results in Table 2. We agree with you very much. According to your comments, we have adjusted the position of "Figure 2" to make the article more rigorous.

  1. Table 1, the percentage for the subgrouping by tumor size (39% and 54.9%, 61.8% and 34.9%) don’t not add up to 100%. It needs to be fixed.

Response:

Thanks for the reviewer’s meaningful criticism. We apologize for the mistakes in the manuscript and also carefully checked the entire manuscript for similar errors.

Reviewer 3 Report

This is interesting paper. The authors assessed the role of adjuvant chemotherapy in breast adenoid cystic carcinoma. The group from SEER in impressive.

The authors should add some information about therapy: type of adjuvant chemotherapy and discuss if old drugs were used.

There should be indicated some disadvantages: HER2 status known only in 36% of patients. 

The number of pts at stage IIb/III is small so the calculation have no power to indicate the role of adjuvant chemotherapy in this group - it shoud be mentioned. The authors state that this indicate no improvement  - based on that small group there is no possibilility to have that conclusion. Also the number of patients for subgroup analyses are too small to get some strong recommendations.

It would be nice to propose some recommendations for clinicians how to treat ACC TNBC - to recommend neoadjuvant chemo at stage II or treat in other way?  

Author Response

Reviewer response

Comments and Suggestions for Authors

This is interesting paper. The authors assessed the role of adjuvant chemotherapy in breast adenoid cystic carcinoma. The group from SEER in impressive.

Response:

We appreciate your positive evaluation. Your comments were highly insightful and enabled us to greatly improve the quality of our manuscript.

The authors should add some information about therapy: type of adjuvant chemotherapy and discuss if old drugs were used.

Response:

Thanks for your meaningful criticism. We agree with your comments and have made corresponding changes in the revised manuscript.

Line 45/Page 9 "In addition, specific chemotherapy regimens, which may have influenced the OS and BCSS, were not available from SEER. However, according to data from the National Cancer Center of China, patients with ACCB received chemotherapy regimens based on anthracyclines or taxanes. Future studies are needed to validate our results and further explore the chemotherapeutic sensitivity of ACCB to optimize the treatment of this rare disease."

There should be indicated some disadvantages: HER2 status known only in 36% of patients.

Response:

Thanks for your meaningful criticism. Unfortunately, this is a disadvantage because some data is missing from the SEER database, which was a pity.

As you suggested, we have re-written the limitations in the revised manuscript.  Thanks again.

The number of pts at stage IIb/III is small so the calculation have no power to indicate the role of adjuvant chemotherapy in this group - it shoud be mentioned. The authors state that this indicate no improvement  - based on that small group there is no possibilility to have that conclusion. Also the number of patients for subgroup analyses are too small to get some strong recommendations.

Response:

Thank you for raising this point. Some subgroups are indeed insufficient to draw corresponding conclusions because the sample size is too small. We agree with your comments and have made corresponding changes in the revised manuscript.

Line 13/Page 9 " It should be emphasized that in some subgroup analyses, such as the stage IIb/III sub-group (only 28 patients), the statistical power is still lacking due to the small number of samples. However, we first proposed the clinical impact of adjuvant chemotherapy in ACCB, providing additional evidence for the management of this rare malignancy. Furthermore, a larger sample size of data is needed for validation in the future."

It would be nice to propose some recommendations for clinicians how to treat ACC TNBC - to recommend neoadjuvant chemo at stage II or treat in other way? 

Response:

Thank you for your guidance here. Based on this, we have included further detailed content in the Discussion section of the revised manuscript. Revisions in the text are shown using the "TRACK CHANGES" mode.

Line 10/Page 9 "Therefore, for patients with ACCB, including tripple-negative ACCB, surgery and postoperative radiotherapy are the preferred treatment methods, and there is currently insufficient evidence-based medical evidence to recommend adjuvant chemotherapy for these patients."
